# Aesthetic and Educational Aspects of Contact with Contemporary Religious Architecture

**Małgorzata Maria Kulik [1], Halina Rutyna [2], Małgorzata Steć [3,*] and Anna Wendołowska [4]**

1  Department of Social Sciences, Institute of Psychology, University of Szczecin, 70-451 Szczecin, Poland; malgorzata.kulik@usz.edu.pl
2  Faculty of Architecture, West Pomeranian University of Technology, 71-210 Szczecin, Poland; halina.rutyna@zut.edu.pl
3  Institute of Psychology, Jesuit University Ignatianum, 31-501 Kraków, Poland
4  Institute of Psychology, Jagiellonian University, 30-060 Kraków, Poland; anna.wendolowska@doctoral.uj.edu.pl
*  Correspondence: malgorzata.stec@ignatianum.edu.pl

**Abstract:** This article addresses the issue of the importance of contemporary architecture—especially Christian architecture—for the aesthetic and spiritual development of an individual. It also highlights the educational aspect that may arise in the framework of the contact of a human with the works of religious architecture. Among many things, the article points out the values of truth and beauty in the space of the *sacrum*. The major importance in the process of human development involves personal, individual and group experiences of meetings in various areas of religious architecture that operate with the language of signs and symbols, modern artistic forms, single-space harmony, and atmosphere—an invisible order of things. In recent years, a number of studies have been carried out that attempted to define what makes the place of *sacrum* sufficiently meaningful, mysterious, and still necessary in order to establish a spiritual relationship with the community of believers and with God, which is relevant in one's transition to adulthood.

**Keywords:** contemporary religious architecture; religious building; sacrum; beauty; truth; goodness; religious development

## 1. Introduction

This article takes the form of a reflection on the relationship between humans and architecture in their perceptual experience. It addresses the question of seeking the values of truth and beauty in the contemporary sacral space and their relationship with the spiritual development of an individual. Architectural and urban solutions of Christian temples may influence or inhibit human spiritual development (Welter and Geddes 2002). Spiritual development is understood here as an important aspect of personal development, as is moral development (Steć and Kulik 2021). Aesthetic development and the related spiritual development can be supported through contact with art, including the art of architecture. Moreover, architecture, being present in the space of every person's life, can have the greatest impact on their sensitivity. Famous British art theorist, Herbert Read, already mentioned the importance of communing with art, including architecture, from the point of view of the overall development of an individual (Keel 1969). This seems to be especially clear in the case of sacral architecture, which is intended to emphasize the values of faith. It can therefore be considered as a tool to support the spiritual development of an individual.

The specificity of contemporary theory and practice of religious architecture is characterized by the language of modern architecture, which is open to the directions of the newest culture, and which reflects the contemporary interpretation of religion. It most willingly speaks through the dynamics of the structure, and the expression of the outer form and is not always in harmony with the traditional canon of beauty (Vidiella 2012). It

benefits from philosophical determinants (sign, symbol, and metaphor), combining them with modern material determinants (structure, natural and artificial light) and aesthetic determinants (architectural structure, plan, spatial layout, design, and ambiance of the interior, compliance with the urban context). All of these aspects can have a significant impact on designing an impact on an individual's impression and thus on their spiritual development.

Each religion has its values and manifests itself with a form of religious architecture placed in the context of urbanized space (Yilmazer and Acun 2018; Aulet and Vidal 2018). Today's architecture is characterized by the plurality of forms and diversity of spatial solutions. This is the result of creative freedom offered by investors and modern technology. The leaders of various Christian communities, leaving the creators free and completely abandoning the use of rigid schemes, canons, or patterns, are also part of this pattern of change. This state of affairs proves that the spirit of contemporary times is characterized by the author's opinions, which require individual reflection. As a result of such wide-ranging limits of creative freedom, the sacral architecture again searches for its spiritual identity and forms of dialogue with the world, using the language understood as the modern language of architecture and art, constantly adapting to the changes and new challenges of the modern world (Aulet and Vidal 2018).

It seems that contemporary sacred architecture is also more aware of its psychological impact on the individual receiving the values that it wants to convey.

The main objective of this study is to explore how modern religious architecture, using the example of Christian architecture, influences an individual in terms of their spiritual self-development. The specific objectives could include an indication of the inalienable role played by the aesthetic factor and the educational aspect in the contact of humans with a modern Christian temple, its urban composition, and its landscape context.

This study adopts a method of analyzing the literature on the subject, analyzing references to the theory of religious architecture, urban planning, and the landscape related to aesthetics. Case studies and analyses performed in their location (in situ) and the evaluation process in the comparative summaries are supported by theoretical studies. The text is interdisciplinary and mainly uses the achievements of the theory of architecture and elements of religious studies, philosophy, psychology, and educational sciences. Thanks to this approach, it is possible to show the multidimensionality of the functions performed by contemporary religious architecture, especially in the context of the spiritual growth of an individual. Sacred architecture can sensitize an individual in terms of faith. It embodies the values of the religion it represents.

It is assumed that contemporary religious architecture can perform many functions, including supporting the spiritual development of an individual, which may translate into their mental health. Thanks to the appropriate construction of the sacred space, contemporary religious architecture can promote a community, as well as the individuality of meeting with God. Through the means of expression used by contemporary religious architecture, it is possible to obtain many different effects that correspond to the needs of contemporary religious believers. Meeting their needs, including spiritual needs, is a significant contribution of architecture to the promotion of mental health alongside the fulfillment of other functions.

In this study, the name religious architecture refers to the art and design of religious buildings. It is being used as the most general term for describing and explaining the role of a sacred place of religious meaning. In a more specific sense, the term religious building can be used to describe a temple, shrine, or church. Each of the aforementioned buildings is a sacred place and is therefore a religious building representing religious architecture.

## 2. Beauty and Truth in Religious Architecture in the Process of Fostering Spiritual Development—The Educational Aspect of Contact with Religious Architecture

The values of truth, goodness, and beauty are considered to be the transcendentals, according to the view prevailing in classical Christian metaphysics that every being is consistent, knowable, and located in the structure of reality. Transcendentals are special

values shared by all entities. They are properties of being and seem to be transferable to each other. Beauty presupposes truth and truth presupposes beauty, especially if these values are justified by God. Recognizing truth, goodness, and beauty as transcendentals comes from the Christian concept of reality as God's work and the embodiment of God's will. In this approach, all three values are interrelated. One cannot speak on any of them without taking into account the others. The concept of transcendentals was explored, e.g., by Thomas Aquinas and Albertus Magnus (Aertsen 1996).

Research on beauty and the issues of aesthetics has been conducted since the beginnings of human thought. The architecture was characterized by the desire for beauty, which was linked to spirituality and, at the same time, to a high level of moral development (Welter and Geddes 2002). According to ancient Greeks, the concept of good and beauty was inextricably linked (Plato, Aristotle), which was also a kind of principle, an idea that was highly endorsed and even desirable (Rzevski and Syngellakis 2018). For the ancient Greeks, the link between good and beauty was referred to as *kalokagathos* (Preus 2015) and was also presented as a model for civic education (*paideia*). This can be considered as the beginning of intuition, saying that shaping a person is not just about a moral aspect but also an aesthetic one, and maybe even more, as through the aesthetic influence one can make spiritual and personal changes in the individual. The old intuition is today confirmed by scientific research, which shows the correlation between activation of very similar brain areas during moral and aesthetic judgment, for example, the medial orbitofrontal cortex and ventromedial prefrontal cortex (Dietrich and Knieper 2021). This may indicate that the human reactions towards something beautiful and something morally desirable can be very similar in terms of neurobiology. Perhaps it is about mediating both situations through strong emotional involvement, which in turn must involve subcortical processing at the same time as the reaction of cortical areas.

Philosophical reflection on architecture is a reflection on the essence and the way architecture exists as a work of art and a work of construction embedded in the world of humans (Michalski 2015). The aim is to design and develop space aesthetically. The beauty in architecture is a positive aesthetic property resulting from the preservation of good proportions, the harmony of colors and sounds, appropriateness, moderation, and utility, perceived by the senses. According to Hegel, beauty is not only connected to balance as a sole result, but also to the tension that leads to this result. The harmony which, as a result, negates the tension that is created in it, changes into something disturbing, false, or even dissonant (Najdek 2021). Therefore, sacred architecture is related to the value of beauty in a special way. The beauty it represents is not only meant to serve humans and meet their subjective needs but also to be a testimony of the divinity itself. Thus, beauty in sacred architecture eludes its particular interpretations and refers to a higher, more objective order, e.g., harmony. In this way, beauty corresponds to the value of truth (Battaino 2020). The more objective it is, the truer it is. At this point, one can speak of true beauty that is intersubjective, indisputable, and divine.

In the theory of classic architecture, there are concepts and principles that are considered important in achieving the harmony and order of the structures, which serve the value of truth. It emphasizes the balance between different parts of a structure, symmetry, the principle of "golden" proportions, and the concept of purpose. The maintenance of a good proportion is the principal property on which architectural composition is based and is understood as a system of interdependencies between certain parts, which exposes the importance of a building (Monestiroli 2004). Just as the work of God is harmonious, the architecture devoted to him is to emphasize this harmony. In the universe exists a special combination of shapes, numbers, and patterns. It embodies the golden ratio. Its existence is a testimony to the creator, God himself (Willson 2003).

According to Husserl (Hajdamowicz 2021), the idea of architectural concepts is a self-contained being, which provides a timeless dimension to all ideas born in the consciousness of an artist. The architectural structure of a sacral object usually attempts to present high aesthetic values and to induce, in the process of perception, specific references to

the knowledge resources of the recipient and influence moral attitudes and emotional feelings. Heidegger claimed that architectural work is not an abstract organization of space. Architecture is where the truth is present, where the artist is the first witness of this phenomenon (Michalski 2015).

However, Ingarden believed that the work of art is built out of a neutral framework of artistic values, which was overbuilt with aesthetic values, constituting an aesthetic-specific realization of the work (Ingarden 1966). He considered artistic values as tools, ancillary means in the face of aesthetic values, allowing these values to appear. However, aesthetic values are derived from artistic values. While artistic values are the direct result of the work of an artist, the aesthetic values for their existence and the assessment of artistic quality (beauty) require active interaction between the subjective perception of a co-constitutive aesthetic object (Stróżewski 2002). Similarly, Stanisław Ossowski shared the aesthetic assessment of the work, giving a psychological interpretation to the term of beauty, by delamination into the values of objective beauty, attributed to objects and artifacts, and of subjective beauty (every recipient has the right to assess because these values are democratic) (Ossowski 1958).

The relationship of the *sacrum* with beauty is symbiotic. Gołaszewska describes this relationship as the "aesthetics of spirituality". She believes that the uniqueness of religious architecture consists of inclusion in this term as the result of a strong emotional interaction, related to the stimulation of religious feelings. This interaction is implemented through several artistic expressions, ranging from the location and composition of the whole work, as well as its parts, to details and material (Gołaszewska 2001). In religious architecture, the desire for beauty is expressed in the creation of forms that affect the viewer, which exacerbates the perception of the user of space in a multilateral manner. Religious architecture is not a flat perception of an image, in which the essence is merely the visual interaction of forms, colors, illusions, or associations. According to Gołaszewska, religious architecture resonates with the three-dimensionality of space along with the entire spectrum of multi-sensory effects through spatial attributes such as scale, rhythm, articulation, shapes, lighting, detail, interior decoration, acoustic feeling, odors, temperature, and the touch of materials and textures. This immeasurable wealth of sensory experiences, when combined into a single perceptional system, creates and reinforces the uniqueness of the architectural form and its sensory multilateralism (Paszkowski 2017). In turn, according to Wojciech Kosiński (2011), a theoretician and designer of the new religious architecture, it is the sacred building that is the highest form of architectural work, which evokes feelings of high religious and aesthetic exaltation, combining mysticism and beauty, which in many examples takes unimaginably strongly influencing forms. The feelings evoked by an architectural form can become the beginning of a process of spiritual ascent, which is part of an individual's spiritual development and therefore can be an important aspect of their mental health.

Today, it is difficult to clearly and objectively define the beauty in architecture and it is difficult to identify the important values and the path that leads to it. In theory and practice, subjective beauty is easier to define because it is based on individual perception and simple analysis of examples and subjective judgment. Identifying objective beauty in architecture, as well as its basic artistic values and ways of achieving a positive result in the theory and methodology of architectural and urban design is difficult, which, as a result, raises many questions during research on beauty and when new work is created. Especially in the context of religious architecture, aesthetics should be accompanied by ethics and thus a return to the ancient definition of beauty. Ultimately, what is beautiful can be considered holy if it is given religious values. In Christian metaphysics, beauty plays a special role. It underlines the greatness of God and testifies to the truth about him. It is also meant to lead to an experience of the sublime. What is holy can also become beautiful in its way, because it is exceptional and recognized as holy by the local religious community. *Sacrum* in the presence of beauty becomes the opening of a mind and soul to the perception of this beauty, which may also involve a certain definition of the value of truth as beauty is

closely related to truth and good (Paszkowski 2017). Communing with both values can have a significant impact on the process of spiritual development of an individual.

If we take into account the possibility of transferring the values of beauty, truth, and good through architectural forms, it can be assumed that the architectural form can become an inspiration for a believer, which refers strongly to self-development. In this aspect, the educational role of religious architecture is implemented. It can be expected that a believer, participating in the rites of its religion in characteristic temples, has the possibility of not only realizing the cult itself or being in contact with the *sacrum* but also of experiencing certain aesthetic feelings that may support its both spiritual and moral development. We already know that aesthetic development goes hand in hand with moral development as the value of beauty is very close to the value of the good (Paris 2019). Both are crucial in the personal development of an individual and may contribute to maintaining mental health. Personal development, striving for subjective autonomy is inextricably linked with moral and spiritual development. Someone developed as a person represents a high moral and spiritual development potential at the same time. The indicators of such development will be high sensitivity and the ability to perceive values and act in accordance with them. According to Stefan Szuman (Polak 2021), development is about moving from ambivalence to ethics. In nature, man is neither good nor bad. Only as a character does he become good or bad, according to the virtues he has or the lewdness he is subjected to. Virtues are an acquired, relatively permanent disposition to act in a good and noble way, and the stability of character in doing good is learned from experience. This is why virtuousness is a positive result of internal battles being fought. Among the main virtues of a human, Szuman mentions moderation, love and justice, godliness (faith), courage, integrity (dedication to the truth), and spirituality (nobleness). It seems that the nobleness proposed by masterpieces of sacral architecture can be of educational meaning in the sense that it forms the basis for shaping certain attitudes. Szuman, similarly to Herbert Read (Keel 1969), was a supporter of bringing up art, as well as aesthetic education (Polak 2021). This is one of the most important objectives of modern holistic education.

Other authors have also highlighted the possibility of acting through art, including the impact of architectural space on the development of an individual (Keel 1969; Mesquida and Inocêncio 2016). According to Herbert Read (Keel 1969), contact with works of art and the artistic activity itself is a catalyst for positive changes in many different areas of life and the functioning of an individual. In the case of religious architecture, we would expect to implement not only the Kantian category of elevation (*the sublime*), which goes beyond the familiar concept of beauty and extends it by a more transcendental dimension but also a specific non-verbal message of certain axiological contents (Specht 2013).

People with developed aesthetic sensitivity also usually have a very wide repertoire of moral virtues corresponding to moral values they believe in. A high level of morality can be an indicator and, at the same time, a manifestation of both aesthetic development and the spiritual development that goes hand in hand with it. Being sensitive to aesthetic values may be associated with a higher level of insight, self-reflection, and the ability to read aesthetic values among other values. This deepened level of sensitivity can also influence moral reflection. It has been known for a long time that by supporting aesthetic development, one can also contribute to growth in the field of moral development (Reid 1982). The upbringing through art is, therefore, an upbringing not only in light of certain behaviors or attitudes toward aesthetic values but also in light of moral values. People who are sensitive to art are also sensitive to moral aspects of reality. This is demonstrated by the most recent results of neuroimaging studies, which emphasize the parallelism of activity of certain brain areas when performing moral and aesthetic judgments. The fMRI study has shown that the attractive faces of males were better remembered by female participants. Neuroimaging has shown greater activity in hippocampal and orbitofrontal areas of the brain during the study which may serve as an argument for the significant involvement of prefrontal cortices in the process of an aesthetic assessment (Tsukiura and Cabeza 2011). This indicates that morality and sensitivity to aesthetics share a common biological basis.

This would explain why they are so closely related. This also justifies why by following the value of a good one can simultaneously realize the values of beauty with greater probability and vice versa.

It seems, therefore, that aesthetic values and their embodiment in artifacts can constitute important tools supporting educational and pedagogical processes, both in terms of education with the importance of religious and socio-moral values. This approach is very close to the idea of psychodydactics, which uses psychological knowledge and methods to support teaching and educational processes (Steć and Kulik 2021; Steć et al. 2021). Moreover, recently, more and more interdisciplinary areas of interest for researchers, such as *moral architecture,* are becoming common place (Edginton 1997). Its fundamental objective is to promote the design of space in such a way that it is appropriate to the social and moral context, and thus, for example, that the various social groups are not excluded by certain features of the design (e.g., contemporary, modern, almost sterile interiors of certain public places exclude the use of such spaces by the poor or homeless people, or even sick ones).

Architecture itself can have a positive, moral, inclusive meaning, as well as a negative one, morally reprehensible through excluding, dominating, or overwhelming. A shrine, a place of religious worship, is very vulnerable to the influence of such tendencies. It would be worth taking this into account already in the process of the creation of architectural projects. As a sacred building can attract and deter believers, it can, therefore, provoke specific experiences and associated emotional states, which in turn can translate into the persistence of certain attitudes and resulting behaviors (Marti 2009).

The problem of risks in architectural creation is raised by Paszkowski in the context of the architect's desire for beauty. Architectural creation can be largely affected by serious threats and the rural shrine of Our Lady in Licheń Stary, the largest shrine in Poland, is given as an example. Despite the use of a classical canon of beauty, the architecture of the building raises serious concerns (Paszkowski 2017); there is a question of the inappropriateness of the form of the shrine to modern aesthetic needs and challenges, the architecture of which copies the canons of the counter-reformation Baroque church. The scale and construction of the shrine might be more of a symbol of the triumphalism of the contemporary Catholic Church than a place of prayer and contact with God (Rutyna 2007).

The composition of architecture and the interior design may give rise to calming our anxiety, comfort or discomfort, a sense of harmony and order, or a feeling of being bored by the excessive number of stimuli. Therefore, in the design of the sacral religious architecture, the "less is more" principle, proposed by the master of minimalism in architecture Mies van der Rohe, is true (Mycielski 2020). On the other hand, new religious buildings with a minimalist form, so eagerly promoted in the industry's architectural journals, which are most popular among critics and architects (Bertoni 2002), may be rejected by the believers due to the overexposed space in their interiors. The emptiness is a kind of synonym for nothingness, and nothingness stands in contradiction with the main message of Christianity and takes away the spirit of uniqueness and holiness, which is not conducive to spiritual growth.

## 3. The Role of Architectural Form and Sacred Interiors in Promoting the Climate of Spirituality

Rituals play a role in building social cohesion and a collective identity that promotes a commitment to collaboration, trust, and tolerance (Whitehouse et al. 2012). Religious spaces are not only where a ritual takes place, but also give additional meaning to ritual practices (Kilde 2008). The aesthetic experience resulting from communing with architecture is similar to the experience of spirituality (Grimm 2010). Art changes our perception of the world and gives meaning to future experiences (Jackson 1998) that are preserved as an integral part of the self (Dewey 1934). As a result of contemplation and spirituality, awe (Hu et al. 2018; Kearns and Tyler 2020) and enthrallment are born (Stebbins 2015). Awe has been related to reduced negative effect (Lopes et al. 2020), enhanced well-being and life satisfaction (Anderson et al. 2018; Dong and Ni 2020; Rudd et al. 2012), pro-sociality

(Sturm et al. 2020), and pro-environmental behavior (Wang and Lyu 2019). It was also identified as a primary component of conversions and religious experiences (Keltner and Haidt 2003), and a starting point for religious development (Negami and Ellard 2021). The awe and wonder experience is associated with a feeling of deep humility resulting from the majesty of God and our personal insignificance, which can result in fear and increased connectedness with other people, but also can lead us to a sense of transcendence (Negami and Ellard 2021; Sturm et al. 2020).

Religious buildings are considered to be places where a human can meet transcendence or be in contact with spiritual reality (Wiśniewski 2018). The sacral space is an area of expression of human ties with God, a place of collective and individual prayer of a human, and an area of influence where one can read the meanings given by the designer, received by the knowledge and intuition of the observer, who is the user of the sacral building, and indicating the values of beauty and truth, because sacral architecture is supposed to illustrate the invisible order of things (Bermudez 2015).

From the very beginning, humans have felt the existence of a transcendental being, which, by its unrecognizability and inexpressibility (i.e., its secret), fueled a sense of fear and terror, overwhelmed with power, but at the same time attracted and stunned. In societies discovering this being, there has been a need to create circumstances conducive to the encounter by, among many things, organizing space and arranging holy places. Sacred places are still needed in the modern era (Harries 2009), and their main function is the meeting of God and humans (Gieselmann 1972).

The typology of the contemporary holy place is the result of the evolution that took place over the centuries and bears traces of successive breakthroughs, linking the history of meetings of human communities with God with traces of the process of revelation of the *sacrum* in the history of humanity. Architecture, spirituality, and contemporary culture and spirituality can have a positive impact on human life (Bermudez 2015).

There are different models of religious buildings characterized by the content of architecture and its order of things, which is appropriate to the arrangement of the layout of a holy place. Subsequently, it is important to have the right atmosphere of the place, directed in the arrangement of the sacral interior, the size of the building, and the utility and synergy of the newly designed building with the urban environment. The characteristics of the five categories of architecture (ideological and significative content, atmosphere of architecture, form of the work, function of the work, and urban planning and environment) are suitable to three theoretical models of contemporary religious architecture, often supported by the tradition of a given fraction of Christianity, whether Eastern or Western.

The first model of a Christian temple, the House of God, is willingly used in the design of contemporary Orthodox churches in which the category of focus is exposed (Lanzi and Lanzi 2005). Its interior is dark and mysterious. The center is emphasized by canonical iconostasis and a round-shaped large chandelier. In this type of solution, the spatial layout of the function of a sacral building is secondary. The investors/leaders of Eastern churches require that it complies with the canon of religious tradition, being transparent and uncomplicated. However, in the design of the interiors of Catholic sanctuaries, an important design problem is the organization of wide internal communication lines, ensuring smooth and collision-free flows of large groups of pilgrims in both directions (entering and leaving the place of worship). In the case of a sanctuary, there is a traditional indication that the atmosphere of its architecture is mystical and humanized, and that the form of architectural structures is original and even iconic. In this case, the structure of a temple constitutes a dominant or at least a landscape accent in an urban context. In this way, it seems obvious and supported by a centuries-old tradition to ensure the visibility of the architectural religious symbol in public space.

The next spatial model, the community house of prayer, is a reasonably organized space of meetings of a small liturgical group that meets the contemporary needs of small and medium-sized Catholic parish, monastic, and secular communities or Protestant churches (Girgis 2011). The atmosphere of such a place is characterized by its intimacy and coziness,

and its form is characterized by modesty and simplicity. The urban context seems neutral because the unremarkable architectural structure does not require visibility or religious manifestation in public space.

There is a growing interest in the early Christian writings of ascetics and hermits. The expression of this interest in contemporary sacral architecture is the return to ascetic, even purist, aesthetics and the simplest, even modest, yet unambiguous forms of early Christian structures. However, the present problem is the quality of the minimalist architecture and how it is received by the public (Podhalanski 2019).

The church meeting room is a large venue for the assembly of a larger community of believers. This type became popular during the baroque period in the purist Protestant temples, especially Lutheran ones, in large Catholic parish churches. It was particularly successful in the solutions of pilgrimage sanctuaries from the Rococo period (e.g., in Bavaria). This model of architecture is reflected in the considerations of Anderson and Sternberg (2020), who believe that the beginnings and changes of the modern *sacrum* have a decisive influence on the spatial design of buildings destined for modern worship and cultural modernity. In this case, the atmosphere of architecture is considered secondary, as the well-established spatial arrangement of a large, well-lit, warm, and ventilated room intended for a crowded liturgical assembly becomes primary (Bergamo 1994). As a result, modern religious architecture designers do not focus on ideological issues and the importance of interacting with the architecture of meanings, but on the existing comfort standards in public utility buildings, by applying the latest technology in terms of acoustic, visual, comfort, and air exchange solutions, among others.

In search of the contemporary canon of Christian architecture, a project path can be found, which leads to an open architectural structure (Bogdanović 2017). Its value is reflected in the synchronization of several levels of architectural work: ideological and religious aspects of the designed churches with a utilitarian layer and a sphere of formal expression of a building. With the meaning layer, an architectural form is the result of an influence of various factors and references, authors' interpretations, subjective thoughts, and professional experience. On the other hand, in the utilitarian layer, linked to the issue of building a material structure, an architect seeks to find the right spatial arrangement, appropriate to the needs of a particular community of believers. Both aspects (layers) constitute the final image of the shaped sacral object, complement each other, and interfere with each other. For a human staying in sacral space, they are to help focus, i.e., to concentrate attention during a meeting with himself and with transcendence.

Only the selected architectural features are conducive to the experience of God; they are often present only in the first model of the perfect church through the symbolism of the structure and detail. In this case, the main category of the *sacrum* is emphasized, which defines the primary and timeless function of the temple, pushing the remaining functions into the background.

The tool allowing us to create a universal environment, i.e., sanctuary, is the canon of Christian signs and symbols (Baldock 1994; Rutyna 2007). The structure and plan of the church itself should show the real and symbolic significance of the sign of the assembly of the believers (Durandus 2020).

The symbolism appears together with the definition of the shape of Christian liturgy, which is followed by the choice of the form of places of a liturgical assembly, or a church building (from the fourth century). Its elements persist over centuries, albeit processed according to the evolving shape of liturgy and the para-liturgical elements, depending on the type of religion embedded in a particular environment and finally on the stylistic evolution (Eliade et al. 1974). The rise of Christian symbolism falls on the period of construction of cathedrals (11th–15th century). Medieval thought was permeated by the conviction about the symbolic nature of the sensual world (Eliade 2004). The world of objects was only real through the symbol (Hani and Quingles 2000) in a symbolic world created by the desire to combine natural and supernatural things, in a game of different configurations (Eco 2009). Symbolic thinking was structured in two ways: it organized

reality, which was not always in line with the doctrine, and at the same time, using the same signs, it captured the same doctrinal truths that might seem inaccessible in scholarly form (Krenz 1997).

Ontological and psychological participation in the symbol turns out to be important, as the symbol serves as the communication of the recipient with sensual forms based on similarities (Forstner 1990). In this case, it is necessary to be able to open the dimensions of the reality that, without it, would have remained closed. The symbol refers to the emotionality of a person. It is polysemous and multi-layered (Eliade 2004). Its nature is by definition metaphorical and ambiguous (Grześkowiak 1976). Therefore, it belongs to the spiritual order, as an intuitive and semi-intuitive feeling (e.g., in the scope of religious feelings).

The light and textures of interior surfaces play a special role in interior design. The lofty, rough interior features the mood of the architecture of Zumthor (Figure 1a), who ensured an atmosphere of a subtle play of silence and light. Each building is created with a particular function in mind, at a specific location, and for specific use (Boyer 2021). Zumthor places the human perception of space at the heart and, thus, extends the humanistic aspect of architectural and urban design theory and design, as well as underlines the sensational aspect of architectural perception (Stec 2020). Definitions of the atmosphere and methodologies of its description are increasingly entering the area of neurophysiology and psychology, in addition to value contemplation, literary descriptions, photographs, and scenery (Figure 1b). The materials used in the building form an architectural symphony, where the perception does not only take place not only with the sense of sight but also with touch, smell, and sometimes taste–a modest but meaningful significance of the place. In this light, he implements the program of historical intellectual and graphic reflection by description, photographs, and scenery.

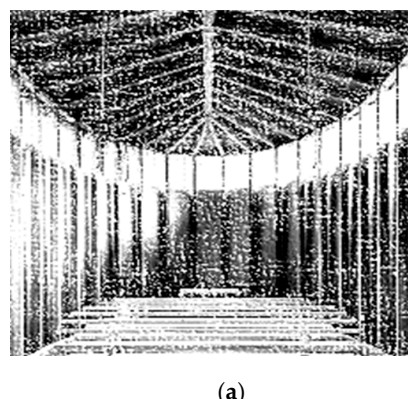    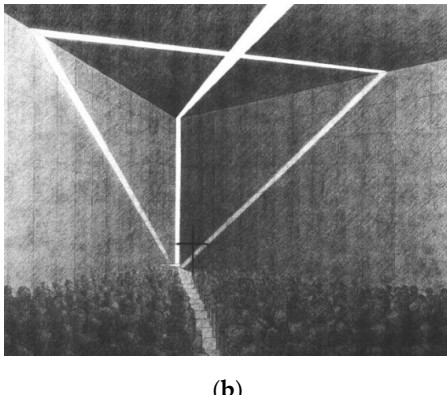

(**a**)                                                  (**b**)

**Figure 1.** The special role of light in the arrangement of sacred interiors: (**a**) Sumvitg (Switzerland) Chapel Sain Benedict by Peter Zumtor, 1988; source: author's graphic design; (**b**) Ibaraki Circle Osaka (Japan) Church of Light by Tadao Ando, 1999; concept and implementation. Source: author's graphic design.

The symbolic choice of colors and light played a special symbolic role in Gaudi's works The Portal of Hope was supposed to be green; the eastern façade presenting the mysterious joyful life of a hidden Holy Family would be brightened by a lot of colors, while the façade of Glory, to which the broad stairs led, inviting to the Kingdom of Glory, would be partly gold-plated (Zerbst 2002). In the urban symbolism, the 140-m tower of Jesus dominating the city was supposed to be illuminated, while the cross crowning the tower's head would cast a strong beam of light on the whole city. In Copenhagen, Peter Klint demonstrated the search for perfect harmony referring to music, included in an architectural play of structures and planes, as well as a clear longing for the Gothic. A person present in this temple does not doubt that he is in contact with a contemporary work of architecture, which creatively takes on the heritage of the European Gothic and the thread of the Danish national tradition (Pallister 2015).

In Paris, the Sacré-Coeur (the Sacred Heart of Jesus) Basilica was built in Montmartre according to the design of Paul Abadie and Lucien Magne (Loyer 1987). The rich and colorful interiors of the temple are decorated with mosaics and marble trims, which is a perfect continuation of Byzantine design. In the design of symbols of the urban concept, which speaks the language of *sacrum*, the urban values of the temple were emphasized as a sign of the biblical "city on the mountain", taken from the Middle Ages. High stairs with viewing terraces lead to the Montmartre hill, creating together with the church a monumental urban setting, one of the emblematic panoramic accents that dominate the city (Norberg-Schulz 1974).

The interpretations of the most recent *sacrum* space may vary in many respects because they depend on the extent of the accepted ideological and theoretical aspects of the work, as well as the conventions, that are chosen for its expression, i.e., the author's inspiration (Nebelsick et al. 2017). Lastly, the depth of influence of the *sacrum* sphere is independent of the choice of location and context of the place, the neighborhood of urban values, the form and function of the sacral object, the atmosphere of architecture, and the ambiance of its interior. The keys to creating this experience and the positive reception of architecture and sacral art buildings are signs, symbols, and metaphors directly based on spiritual content. The interior design, icons, and symbols used are intended to create an appropriate architectural atmosphere and to emphasize the importance of space and the spiritual meaning of religious rites (Hills 2018).

## 4. The Importance of Urban Qualities in the Choice of Location and Context of the Place of Sacral Architecture

For centuries, humans have shaped the space of their lives according to their knowledge and faith, giving many important attributes linked to tradition and culture to a given place. Humans created housing and work environments, as well as religious needs, while also entering the natural and cultural landscape. Architecture and urban planning have always been and continue to be the background for all human activities, the plane of shaping social life and the spiritual development of an individual. The human shapes their housing environment—the space of human life—with time models and shapes their own life (Paprzyca 2012). In its urban development plan, the historic city's urbanism provided for a reservation of a separate parcel in each district of the city, intended for the construction of a sacral district. Urban planners defined its dominant structure well in advance, and sometimes its aesthetics about the environment. The 20th century brought a completely different scale and design of urban residential and sacral buildings, and the 21st century moved the aesthetic problems of urban space in practice into the background (Paszkowski 2017).

The structure of the new architecture becomes part of the existing space and landscape. It is, therefore, important to respect the context of the site and to fit the new building into the existing environment and landscape—both natural and built-up (Wrana and Fitta-Spelina 2017).

The presence of religion in cities is realized through architecture. The choice of location for a new investment by the sacral team is closely linked to the paradigm of "New Visibility of Religion" (Deibl 2020). The depth of influence of the *sacrum* sphere is dependent on the choice of location and context of the site, as well as the urban values of the previously formed sacral environment of the building, which will allow the shaping of significant contact with the new sacral architecture by the users and observers of the sacral space, in occasional or systematic contact.

Sacral structures play an important role in shaping the public space of cities and villages in determining their composition relationships with the environment and in defining a clear and understandable language of symbolic forms of architecture. Their monumental structures often become icons of the city and signposts in the town planning system (Jencks 1973). On the architectural scale, the structures and interiors of modern churches offer many new possibilities for guiding users of space toward mystery. A believer, who

crosses the threshold of a church, has the chance to enter the space where actual contact with the transcendence may and should occur (Rutyna 2007).

*Challenges of Contemporary Architecture in the Context of Promoting Spiritual Development*

The new form of religious architecture has the potential to introduce a modern human into the living experience of the *sacrum* and should respond to the current religious and aesthetic needs of the communities of believers (Arweck and Keenan 2016). The general criteria for the assessment and classification of religious architectural works (because they are objects of general interest, and in terms of general architecture, they are considered as cultural services) should be used in the process of design and evaluation of religious objects.

Today's worldwide examples of sacral objects clearly show the phenomenon of cutting off from the historical patterns of tradition and the creation of new architecture in the dynamics of structures and expression of the form (Warren 2018). As in the original Christianity, sacral structures showed the imbalance between the shape of the interior and the external silhouette, and this is also the case today. Modern architecture speaks mainly through the dynamics of the structure and expression of the external form. At present, this discord is on the opposite pole about structures derived from the time of Christian antiquity, to turn architecture from the outside to toward the structure and to neglect or neutral treatment of internal expression. This may be linked to technological opportunities brought by contemporary times in the scope of architectural design, and perhaps to the feeling of jeopardizing religious values through secular ones. Despite this trend, the simultaneous emphasis on the interior allows spiritualizing new religious buildings.

Sacral architecture may be an art open to the poetics of *sacrum*, which inspires to discover oneself and shape oneself towards maturity. In this case, maturity means being ready to enter an individualized contact with transcendence (Anderson and Sternberg 2020). In this understanding, maturity is also personal and therefore moral maturity. Someone developed as a person is also a highly moral individual, because morality is something that demonstrates having developmental autonomy in terms of being a person, an agent responsible for one's life and its impact on the reality.

Ultimately, the art contained inside a religious building helps or hinders the understanding, performance, and experience of liturgy, may also cover or distort its image, and may compete with the action of the ceremony. It is worth remembering about the interplay of the dynamics of the interior design elements and the perception of a believer, who moves, walks, kneels, or stands up, alternating between static posture and movement. This interplay allows art to form a uniform whole with the dynamics of the rite, with gestures, words, music, and singing (Rutyna 2007).

Contemporary times brought a wide range of means of expression and dynamism of forms in the sacral interiors (Bermudez 2015). Outstanding architects propose the poetics of *sacrum* in the hidden interior of new religious buildings, visible by the luminism, as well as the construction of lighting scenes using artificial light Similar characteristics of the new architecture, which is poor in artistic means of expression in the interiors of other Christian temples are proposed by its creators: Meinhard von Gerkan (Christ Pavilion in Volkenroda Monastery in Germany), Steve Holl (Chapel of Saint Ignatius in Seattle, Washington), Juha Leiviska (Myyrkami Church), and Richard Meier (the Jubilee Church in Rome). In these works, the *sacrum*, which is defined differently as a dedicated, sanctified, and separated space, i.e., different, now, despite the difference in form, in these cases, it has been given an architecture that is well integrated into the environment, which is a distinctive feature of the site.

At the time of the intensifying process of self-awareness of a human who, after rejecting all authorities, searches for their own identity and the meaning of human existence, internal anxiety and instability are also reflected in sacred architecture, although it would seem that it should always express the certainty and durability of faith in God—an undeniable, transcendental authority. Religious sacral architecture requires a new language of conveying meaning by the designer in the design process (Aulet and Vidal 2018).

Reinforced concrete structures of temples, as in the case of Perret's Church of the Virgin Mary in Le Raincy being a "technical functional form", most often prevent the creation of a sacred climate inside it (Latour and Szymski 1985).

On the other hand, an extraordinary form of Notre Dame du Hempt in Ronchamp (Figure 2a,b), created by Le Corbusier, is so irrational and at the same time so suggestive that it would direct the reception of the observer. In this case, the tactics of perception of architecture reduced to the floor plan and four façades are fail; these are rather views, afterimages, and an abundance of expressive detail, which allow the structure to escape any intellectual analysis and each view captures a different image (Anderson and Sternberg 2020).

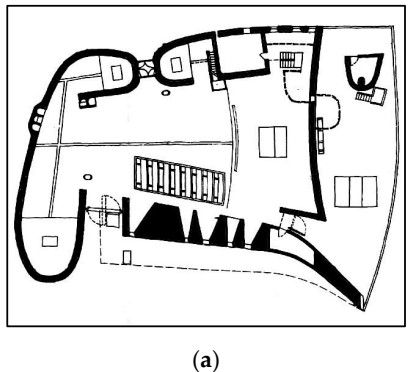
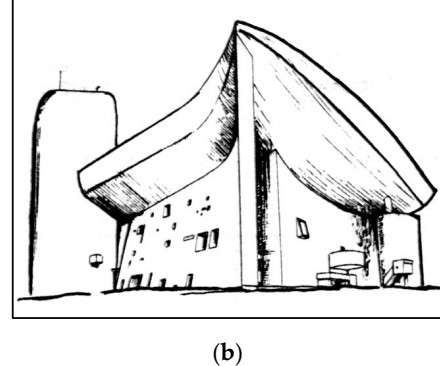

(**a**) (**b**)

**Figure 2.** Ronchamp (France), chapel Notre-Dame-du Haut, Le Corbusier, real. 1950–1955: (**a**) plan chapel; source: author's graphic design; (**b**) external view of the chapel; source: author's graphic design.

Lastly, the St John's abbey church near Collegeville (Figure 3a,b), designed by Breuer and Nerviem, which received a cubic form of "concrete brutalism" (Jencks 1987), harmoniously plays with rhythms of triangles and spatially broken wall planes in its inside. The idea of respecting all the requirements of the liturgical Catholic cult has been perfectly fulfilled in the organization of the interior, with particular regard to the community arrangement of the laity zone. It is focused around an altar, which is centrally positioned and firmly extending toward the center of the church. There is a spatial separation and valorization of the zone of the baptistery and symbolic elements of the temple, such as the tower, gate, and altars.

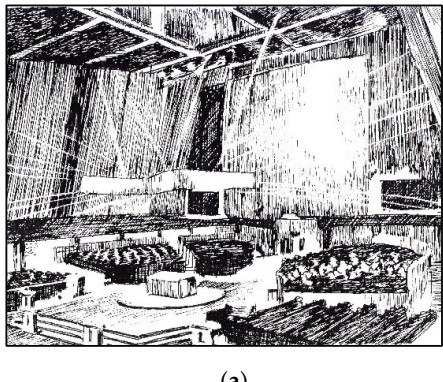
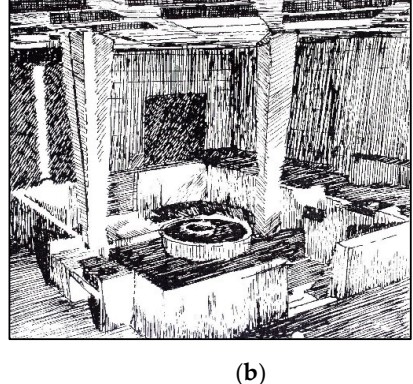

(**a**) (**b**)

**Figure 3.** Abbey, Collegeville (USA), Opactwo benedyktynów St. John's by Marcel Breuer and Pier Luigi Nervi, 1953–1961: (**a**) view of the altar zone; source: author's graphic design; (**b**) view of the interior of the baptistery; source: author's graphic design.

## 5. Conclusions

The reflection on ideological and theoretical issues in the process of supporting the development of an individual by being in contact with sacral architecture has pointed to the inalienable role played by two factors: the aesthetic and educational aspects of human

contact with the modern religious buildings, using the Christian temple as an example. A study of the design of sacral architecture has shown that the relationship of a human with sacral architecture is influenced by many factors.

Modern architecture is characterized by the diversity and plurality of solutions for the form and functions of modern churches. This is linked to the evolution that has taken place in the modern world and in the history of contemporary culture, which affects the new approach of designers to the composition of spatial forms in the 20th and 21st centuries.

In the contact of a human with religious architecture, it is important to take into account the program and religious ritual appropriate to a given faction of the Christian religion. The dynamism of liturgical action and the community nature of worship were highlighted. This has been translated into the currently promoted community arrangement of a contemporary temple, with a marked center. The solutions of the new churches are characterized by diversity and oscillate in all models of the temple. These are most often central structures, which represent the type of the House of God. There are also community houses of prayer which, through a central and longitudinal arrangement, help to shape the communality and unity of the Church. There are also buildings, implemented as large rooms of the church, emphasizing triumphalism and the power of Christian believers through spatial depth and assigning architectural form with attributes of monumentalism. The longing of some creators, and especially the principals, for the ideal of a mystical space, inspired us to turn to the past and to seek the model of a *real* temple, which ultimately resulted in the duplication of patterns and repertoire of forms derived from old, well-proven architectural designs.

After all, the new religious architecture is an art open to new creations. It arranges architectural space in a modern way and uses the means of contemporary art to create an atmosphere of the poetics of *sacrum*, thus inspiring one to look inside oneself. A special place such as a church or a temple builds a space for contemplating the sacred. It allows one to calm down and pray while conveying the impression of God's greatness. All of this can have educational and developmental significance. Just being in the temple can contribute to self-insight and self-reflection. This can be the beginning of self-education work, i.e., an experience of the need for improvement, or compensation. It can form a person towards spiritual maturity. In the mind of the recipient of the sacred space, it creates contents and associations that express and open one to the truth about the beauty and majesty of God. The need for self-improvement in the face of an encounter with God is much more possible in the context of a holy place than outside of it. This makes sacred architecture an exceptionally important mission.

The article points to the inevitable role of an architect and the design decisions they take in a new way. If the creator adopts the rules by which the project will be structured and focuses on one idea, it will allow unity in diversity to appear in a modern artifact. The search for unity, harmony, and spatial order—the main categories of beauty, translates into a potential spiritual integration of the recipient and the user of sacral architecture, being in contact with contemporary architecture and sacral art. The exemplification of ideological and theoretical issues in selected projects of contemporary sacral architecture proves that it is possible to achieve the ideal solution for a new temple or place of prayer in the contemporary space of the residence. An architect who strives to achieve unity focuses on the adoption of one thought, and, as a creator of the work, makes a significant contribution to the creation of good architecture of meanings. Modern humans are sensitive to accuracy, logic, holiness, and truth, i.e., the authenticity of the work. The clarity and legibility of composition are due to the merging of all elements influencing the perception and reception of the recipient of the work of architecture and art placed in the interior of contemporary architecture.

Architecture often emphasizes the importance of beauty in combination with the atmosphere of a place (Birch and Sinclair 2013). As today's world is struggling with climate change, a pandemic, armed conflicts, population growth, economic crisis and the problem

of refugees, it is becoming more and more important to give architectural spaces a deeper meaning that, under the influence of beauty, will evoke inner peace and deep spirituality.

**Author Contributions:** H.R. had the idea for the article and wrote the original draft with support from A.W., M.S. and M.M.K. who performed the literature search and critically revised the work. All authors have approved the submitted version. All authors agree to be personally accountable for the author's own contributions and for ensuring that questions related to the accuracy or integrity of any part of the work, even ones in which the author was not personally involved, are appropriately investigated, resolved, and documented in the literature. All authors have read and agreed to the published version of the manuscript.

**Funding:** This research received no external funding.

**Institutional Review Board Statement:** Not applicable.

**Informed Consent Statement:** Not applicable.

**Data Availability Statement:** Not applicable.

**Conflicts of Interest:** The authors declare no conflict of interest.

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
