# Peer review of "Aesthetic and Educational Aspects of Contact with Contemporary Religious Architecture"

_religions, doi:10.3390/rel13050418_

Round 1
Reviewer 1 Report
Aesthetic and Educational Aspects of Contact with Contemporary Sacral Architecture and Art – Review
1) The article has an interesting topic: it deals with the relationship of the individual to sacred architecture and presupposes that the individual can develop in the perception of architecture as a personality. The article mainly refers to spiritual development, which is regarded as part of a broader form of personality development. Two classical categories of philosophical aesthetics function as signposts: truth and beauty. Both are positively related to personality development. In short, the consideration of sacred architecture takes its starting point from the three motifs of truth, beauty and the contribution to personality development. The article has a specific interest in contemporary sacred architecture, the beginning of which it places at the beginning of the 20th century with the endeavour to leave historicism behind and to engage in new forms of construction. Nevertheless, the authors of the article are also concerned with a recourse to the oldest Christian building forms, which, according to the authors, seem to be still highly relevant today.
The basic conception of the article is certainly its strength: concrete reflections on modern sacred architecture (chapters eleven and twelve) are integrated into a larger philosophical and architectural-theoretical context. In my opinion, this is also where the problem of the text arises. I describe this here in general terms and try to make it more concrete by means of comments on the text.
2) The connection of theoretical considerations on truth, beauty and personality development on the one hand and, on the other hand, the description of concrete examples is not really achieved by the text. The authors refer to six different Christian sacred buildings and interpret them (chapters eleven and twelve). In the corresponding reflections, one can certainly find the question of beauty, but I miss the question of truth and the contributions to the development of personality. Until the end of the text, I don’t really understand how this development is achieved. In the "Final Conclusions", it is said: "The truth and beauty of sacred architecture should be an integral part of the built-up space." (830 et seq.) However, this sounds more like a recommendation for the architects than an analysis of existing buildings one can perceive. A similar question could be linked to the following sentences: "The two dimensions of the paradigm of beauty were highlighted as an immanent feature ordering the anthropogenic reality and the way in which the works of architectural art are felt and perceived, as well as the built-in space and the context of the natural environment are received." (831-834) Where exactly are these two dimensions of beauty emphasised? Finally, what character do the processes of personality development ("thus inspires to get [to] know oneself and to form oneself towards maturity", 859 et seq.) have in relation to the buildings mentioned? In what do these processes manifest themselves and how do they take place? What does it mean to say: "In the mind of the recipient of space, it [the new sacral architecture] creates contents and associations which express and open to the beauty and majesty of God." (860 et seq.) Sacred architecture has always struggled with this very question. In different times, there were different ideas about how to refer to the "beauty and majesty of God". What does this look like in contemporary sacred architecture or, at least, in the examples mentioned?
What is the connection between truth and beauty? The article says: "People with developed aesthetic sense, but also seeking to have contact with the values of art, also usually have a very wide repertoire of moral virtues corresponding to moral values. The upbringing through art is therefore an upbringing not only towards certain behaviours or attitudes towards aesthetic values, but also towards moral values. People who are sensitive to art are also sensitive to moral aspects of reality. "(186-190) However, there are also many counterexamples. For me, the rationale for the connection of truth and beauty is too vague: "This is demonstrated by the most recent results of neuroimaging studies, which emphasize the parallelism of activity of certain coloured and sub-coloured areas, just like performing moral and aesthetic judgements (Tsukiura and Cabeza 2011). "(190-193) The text does not delve into neuro sciences, so that argument is completely unexpected and, what’s more, it is not further developed.
3) I am a great advocate of interdisciplinary research, and I am well aware that not all disciplines involved can always be treated with the same care and expertise. Those who engage in interdisciplinary research make themselves vulnerable to reviewers. Furthermore, a full-length article for a journal can not be expected to cover a topic like an monography. For these very reasons, I think it is important to clarify in the introduction from which point of view one is approaching the topic, where the strengths are to be found and where the article also has its limits. In the concrete case: Are the authors writing from the perspective of theology, religious studies, architectural theory, art history, etc.? This should be clarified in the introduction.
An additional paragraph in the introduction indicating the structure of the text (the sequence of chapters) would make reading much easier.
4) The text repeatedly contains assertions that are not argued in more detail. I give one example, others I have marked in the text by means of comments: "From now on, the authors of reinforced concrete temples have not considered that by highlighting the structure frame, the structure becomes a 'technical functional form', which most often makes it impossible to create a sacrum climate in its interior." (766-769)
5) The article needs good language editing, also with regard to technical terminology. Especially the wide use of the term "temple" should be reflected carefully.
6) How well the article fits in the special issue "Religion and Mental Health: Antecedents and Consequences", must be judged by the special editors. I think that in a revision of the article the concerns of the special issue should be made clearer.
Author Response
1) The article has an interesting topic: it deals with the relationship of the individual to sacred architecture and presupposes that the individual can develop in the perception of architecture as a personality. The article mainly refers to spiritual development, which is regarded as part of a broader form of personality development. Two classical categories of philosophical aesthetics function as signposts: truth and beauty. Both are positively related to personality development. In short, the consideration of sacred architecture takes its starting point from the three motifs of truth, beauty and the contribution to personality development. The article has a specific interest in contemporary sacred architecture, the beginning of which it places at the beginning of the 20th century with the endeavour to leave historicism behind and to engage in new forms of construction. Nevertheless, the authors of the article are also concerned with a recourse to the oldest Christian building forms, which, according to the authors, seem to be still highly relevant today.
We would like to thank the reviewer for appreciating the importance of the issue raised. Reading these words, we feel that this is what we wanted to achieve with our text.
The basic conception of the article is certainly its strength: concrete reflections on modern sacred architecture (chapters eleven and twelve) are integrated into a larger philosophical and architectural-theoretical context. In my opinion, this is also where the problem of the text arises. I describe this here in general terms and try to make it more concrete by means of comments on the text.
We will make every effort to respond to all comments and correct any errors and weaknesses indicated.
2) The connection of theoretical considerations on truth, beauty and personality development on the one hand and, on the other hand, the description of concrete examples is not really achieved by the text. The authors refer to six different Christian sacred buildings and interpret them (chapters eleven and twelve). In the corresponding reflections, one can certainly find the question of beauty, but I miss the question of truth [to emphasize the value of truth, we have added sentences that make this aspect more clearly] and the contributions to the development of personality. Until the end of the text, I don’t really understand how this development is achieved. In the "Final Conclusions", it is said: "The truth and beauty of sacred architecture should be an integral part of the built-up space." (830 et seq.) [we have removed this sentence in order to avoid possible understatement] However, this sounds more like a recommendation for the architects than an analysis of existing buildings one can perceive. A similar question could be linked to the following sentences: "The two dimensions of the paradigm of beauty were highlighted as an immanent feature ordering the anthropogenic reality and the way in which the works of architectural art are felt and perceived, as well as the built-in space and the context of the natural environment are received." (831-834) Where exactly are these two dimensions of beauty emphasised? [we have removed this sentence in order to avoid possible understatement as well] Finally, what character do the processes of personality development ("thus inspires to get [to] know oneself and to form oneself towards maturity", 859 et seq.) have in relation to the buildings mentioned? In what do these processes manifest themselves and how do they take place? What does it mean to say: "In the mind of the recipient of space, it [the new sacral architecture] creates contents and associations which express and open to the beauty and majesty of God." (860 et seq.) Sacred architecture has always struggled with this very question. In different times, there were different ideas about how to refer to the "beauty and majesty of God". What does this look like in contemporary sacred architecture or, at least, in the examples mentioned? [some new text passages have been added and/or changed in order to explain this issue more clearly; we hope now the text is of a better quality]
What is the connection between truth and beauty? The article says: "People with developed aesthetic sense, but also seeking to have contact with the values of art, also usually have a very wide repertoire of moral virtues corresponding to moral values. The upbringing through art is therefore an upbringing not only towards certain behaviours or attitudes towards aesthetic values, but also towards moral values. People who are sensitive to art are also sensitive to moral aspects of reality. "(186-190) However, there are also many counterexamples. For me, the rationale for the connection of truth and beauty is too vague: "This is demonstrated by the most recent results of neuroimaging studies, which emphasize the parallelism of activity of certain coloured and sub-coloured areas, just like performing moral and aesthetic judgements (Tsukiura and Cabeza 2011). "(190-193) The text does not delve into neuro sciences, so that argument is completely unexpected and, what’s more, it is not further developed. [there are some changes introduced to this section; we hope now it is more clear and understandable; we have made the effort to implement all your comments and advice]
3) I am a great advocate of interdisciplinary research, and I am well aware that not all disciplines involved can always be treated with the same care and expertise. Those who engage in interdisciplinary research make themselves vulnerable to reviewers. Furthermore, a full-length article for a journal can not be expected to cover a topic like an monography. For these very reasons, I think it is important to clarify in the introduction from which point of view one is approaching the topic, where the strengths are to be found and where the article also has its limits. In the concrete case: Are the authors writing from the perspective of theology, religious studies, architectural theory, art history, etc.? This should be clarified in the introduction. [we have added one passage of the text in the introduction in order to increase the clarity in this matter]
An additional paragraph in the introduction indicating the structure of the text (the sequence of chapters) would make reading much easier. [It has been included]
4) The text repeatedly contains assertions that are not argued in more detail. I give one example, others I have marked in the text by means of comments: "From now on, the authors of reinforced concrete temples have not considered that by highlighting the structure frame, the structure becomes a 'technical functional form', which most often makes it impossible to create a sacrum climate in its interior." (766-769) [we have tried to avoid this situation; we hope we did our best and the result is satisfactory]
5) The article needs good language editing, also with regard to technical terminology. Especially the wide use of the term "temple" should be reflected carefully. [the term has been checked and replaced where it was necessary]
6) How well the article fits in the special issue "Religion and Mental Health: Antecedents and Consequences", must be judged by the special editors. I think that in a revision of the article the concerns of the special issue should be made clearer. [we have emphasized this aspect a few times in order to make it more visible]
Reviewer 2 Report
Major revisions are necessary in terms of organization of text and proper explication in an English translation. Inclusion of illustrations should be considered.
The subject of the article has validity, but the treatment of the subject is rather convoluted because of the lack of organization within the text. As a result, the main tenets of the thesis are primarily reinforced in the concluding section of the paper, and it is difficult to follow the authors' development of the argument in the body of the article. Strong recommendations are as follows:
- Reconfiguration of the text so there is a section of the paper that focuses on theoretical issues that substantiate the argument.
- The entire text needs to be readdressed so clarity of expression is maintained.
- The expertise of an editor who can readily correct the many grammatical errors and difficulties in expression will be primary.
- The article would be significantly improved by the inclusion of illustrations.
Author Response
Major revisions are necessary in terms of organization of text and proper explication in an English translation. Inclusion of illustrations should be considered [illustrations has been added]
The subject of the article has validity, but the treatment of the subject is rather convoluted because of the lack of organization within the text [the text has been widely reorganized]. As a result, the main tenets of the thesis are primarily reinforced in the concluding section of the paper, and it is difficult to follow the authors' development of the argument in the body of the article. Strong recommendations are as follows:
- Reconfiguration of the text so there is a section of the paper that focuses on theoretical issues that substantiate the argument. [the text has been reorganized as adviced]
- The entire text needs to be readdressed so clarity of expression is maintained.[we think we have done it and it should be more clear now]
- The expertise of an editor who can readily correct the many grammatical errors and difficulties in expression will be primary [the text has been checked by a native speaking professional editor]
- The article would be significantly improved by the inclusion of illustrations [illustrations has been added]
Reviewer 3 Report
This manuscript sets out to assess the importance of architecture for the aesthetic and spiritual development of the individual. A very interesting subject.
I think the author needs to revise the manuscript in order to ground this assessment in academic terms. First, reference previous studies that have articulated quantifiable measures of aesthetic and spiritual personality traits. What factors promote or inhibit the expression of such traits? Then---how might different architectural styles influence these factors?
A lot has been written about the scientific study of human emotion and how the evocation of specific emotions influence cognitive and affective states. There has been an explosion of studies on both awe and wonder (important for this study to distinguish between these two and note how the evocation of awe induces submission while the evocation of wonder elicits no submission, but does elicit "higher order" cognition that makes possible the contemplation of metaphysical causality as well as eliciting greater empathy and compassion). The author might also look at Harvey Whitehouse's studies of how ritual behavior evokes certain cognitive and behavioral tendencies. Why might some architecture evoke analytic (noreligious) thinking while other architecture evokes intuitive (religious) cognitive mechanisms?
Line 96 mentions that scientific research linking aesthetic experience and morality----great-----please expand on this and explain what the causal factor operative here is such that certain kinds of architecture manipulate that factor. Similarly, line 230 mentions a study by Tsukivra & Cabeza---explain this study and, again, explain the causal factors involved.
I thought that lines 238-274 were the best in the article.
Author Response
This manuscript sets out to assess the importance of architecture for the aesthetic and spiritual development of the individual. A very interesting subject.
This is why we have created this manuscript. Your words make us believe our work has its purpose. Thank you so much for your words of encouragement.
I think the author needs to revise the manuscript in order to ground this assessment in academic terms. First, reference previous studies that have articulated quantifiable measures of aesthetic and spiritual personality traits. What factors promote or inhibit the expression of such traits? Then---how might different architectural styles influence these factors?
The main objective of the study presented in the manuscript was to explore how modern religious architecture, using the example of Christian architecture, influences an individual but in terms of their spiritual self-development not in terms of aesthetic and spiritual personality traits. We believe that personality traits are one thing and categories of aesthetics and spirituality are something different. It does not mean that there is no link between them. The text of the manuscript is based on an analysis of literature. We added some possible references that we hope would fill the existing gap:
Grimm, Shae. 2010. Architecture & Spirituality. An Architecture-Centered Aesthetic Experience
Birch, Robert, and Brian R. Sinclair. 2013. Spirituality in Place: Building Connections Between Architecture, Design, and Spiritual Experience. In ARCC Conference Repository. New York: Routledge.
Keltner, Dacher, and Jonathan Haidt. 2003. Approaching Awe, a Moral, Spiritual, and Aesthetic Emotion. Cognition and Emotion 17: 297–314.
A lot has been written about the scientific study of human emotion and how the evocation of specific emotions influences cognitive and affective states. There has been an explosion of studies on both awe and wonder (important for this study to distinguish between these two and note how the evocation of awe induces submission while the evocation of wonder elicits no submission, but does elicit "higher-order" cognition that makes it possible the contemplation of metaphysical causality as well as eliciting greater empathy and compassion). The author might also look at Harvey Whitehouse's studies of how ritual behavior evokes certain cognitive and behavioral tendencies. Why might some architecture evoke analytic (no religious) thinking while other architecture evokes intuitive (religious) cognitive mechanisms?
Thank you for your interesting and inspiring suggestions. Some ideas were included in the text (lines 280-297). The framework we propose is by no means exhaustive and definitive - there are many other related concepts that are not discussed in detail here. They will certainly be considered in the context of future publications.
Line 96 mentions that scientific research linking aesthetic experience and morality----great-----please expand on this and explain what the causal factor operative here is such that certain kinds of architecture manipulate that factor.
Lines 96-103 – suggested change has been added. Highlighted in red.
Similarly, line 230 mentions a study by Tsukivra & Cabeza---explain this study and, again, explain the causal factors involved.
Lines 234 – 238 – suggested change has been added Highlighted in red.
I thought that lines 238-274 were the best in the article.
Thank you so much for this annotation. We appreciate it.
Round 2
Reviewer 1 Report
The article has visibly gained in quality through the revision. Nevertheless, I would like to suggest some changes, which I have indicated as comments in the attached docuent.
The new structuring makes the text much clearer. However, when the text was reorganized, a few sentences were also left in their old place, so that they now appear twice. Where I noticed this, I have marked it in the text.
I wish all the best for the revision and am available for questions.

Author Response
Thank you for all your annotations in the text. We have previously changed different things according to your comments. We have carefully scanned the proposed changes and accepted all of them this time. The agreement of all authors for all suggested changes has been established therefore the process of amendment was finally successful.
Reviewer 2 Report
The reservations I have about this article are the ones that I had submitted in my initial review. The errors have not been substantially remedied.
Author Response
Thank you for your comment. We have made all efforts to avoid previous errors. We hope it is much better now. Our exploration of the text revealed a few imperfections which shouldn't be there anymore. We hope the manuscript is acceptable now after including all suggestions.
Reviewer 3 Report
The revisions do indeed improve the article---but they were simply tacked on to the existing narrative and do not substantially improve the article. The reader does not really learn what factors influence aesthetic and spiritual development and thus how exposure to architecture affects these factors.
I still believe that this topic requires a more thoroughly revised manuscript. Clearly define what aesthetic and spiritual development is. Is this simply therapeutic progress? Is it part of all humans' lives or only some people? What are the main indicators of this development (measurable or at least detectable manifestations)? What factors promote or retard such development? How does interaction with architecture affect these factors? Have other studies examined aspects of such interaction (e.g., the role of emotional experience on cognitive/perceptual/affective processes; the role of body movement in mental health; etc.)?
I am still troubled with lines 75-78 that mention "truth" (seemingly transcendent---transcendent to what?) and how a reader might know what this means or refers to.
Author Response
The revisions do indeed improve the article---but they were simply tacked on to the existing narrative and do not substantially improve the article. The reader does not really learn what factors influence aesthetic and spiritual development and thus how exposure to architecture affects these factors.
This issue has been fixed and explained. We believe that now t is more clear which factors are to be taken into account while establishing the impact of architecture on development.
I still believe that this topic requires a more thoroughly revised manuscript. Clearly define what aesthetic and spiritual development is. Is this simply therapeutic progress? Is it part of all humans' lives or only some people? What are the main indicators of this development (measurable or at least detectable manifestations)? What factors promote or retard such development? How does interaction with architecture affect these factors? Have other studies examined aspects of such interaction (e.g., the role of emotional experience on cognitive/perceptual/affective processes; the role of body movement in mental health; etc.)?
It has been fixed respectively. All additions are highlighted in red.
I am still troubled with lines 75-78 that mention "truth" (seemingly transcendent---transcendent to what?) and how a reader might know what this means or refers to.
Maybe this is a kind of misunderstanding? We have meant transcendentals which are of philosophical meaning. It is not the same with transcendence. Transcendentals are the highest values that are always taken together in terms of the foundation of all other values.